# TOWARDS FASTER AND STABILIZED GAN TRAINING FOR HIGH-FIDELITY FEW-SHOT IMAGE SYNTHESIS

**Bingchen Liu**[1,2]**, Yizhe Zhu**[2]**, Kunpeng Song**[1,2]**, Ahmed Elgammal**[1,2]
[1]Playform - Artrendex Inc., USA
[2]Department of Computer Science, Rutgers University
{bingchen.liu,yizhe.zhu,kunpeng.song}@rutgers.edu
elgammal@artrendex.com

## ABSTRACT

Training Generative Adversarial Networks (GAN) on high-fidelity images usually requires large-scale GPU-clusters and a vast number of training images. In this paper, we study the few-shot image synthesis task for GAN with minimum computing cost. We propose a light-weight GAN structure that gains superior quality on $1024 \times 1024$ resolution. Notably, the model converges from scratch with just a few hours of training on a single RTX-2080 GPU, and has a consistent performance, even with less than 100 training samples. Two technique designs constitute our work, a skip-layer channel-wise excitation module and a self-supervised discriminator trained as a feature-encoder. With thirteen datasets covering a wide variety of image domains [1], we show our model's superior performance compared to the state-of-the-art StyleGAN2, when data and computing budget are limited.

## 1 INTRODUCTION

The fascinating ability to synthesize images using the state-of-the-art (SOTA) Generative Adversarial Networks (GANs) (Goodfellow et al., 2014) display a great potential of GANs for many intriguing real-life applications, such as image translation, photo editing, and artistic creation. However, expensive computing cost and the vast amount of required training data limit these SOTAs in real applications with only small image sets and low computing budgets.

In real-life scenarios, the available samples to train a GAN can be minimal, such as the medical images of a rare disease, a particular celebrity's portrait set, and a specific artist's artworks. Transfer-learning with a pre-trained model (Mo et al., 2020; Wang et al., 2020) is one solution for the lack of training images. Nevertheless, there is no guarantee to find a compatible pre-training dataset. Furthermore, if not, fine-tuning probably leads to even worse performance (Zhao et al., 2020).

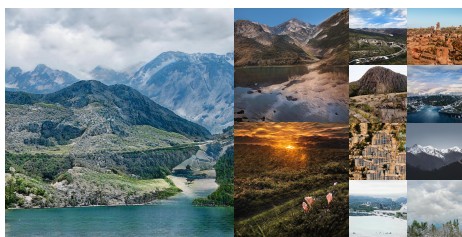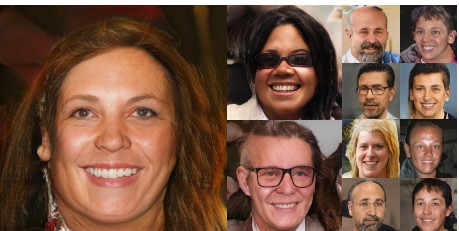

Figure 1: **Synthetic results on** $1024^2$ **resolution** of our model, trained from scratch on single RTX 2080-Ti GPU, with only 1000 images. Left: 20 hours on Nature photos; Right: 10 hours on FFHQ.

In a recent study, it was highlighted that in art creation applications, most artists prefers to train their models from scratch based on their own images to avoid biases from fine-tuned pre-trained model. Moreover, It was shown that in most cases artists want to train their models with datasets of less than

---

[1]The datasets and code are available at: https://github.com/odegeasslbc/FastGAN-pytorch

100 images (Elgammal et al., 2020). Dynamic data-augmentation (Karras et al., 2020a; Zhao et al., 2020) smooths the gap and stabilizes GAN training with fewer images. However, the computing cost from the SOTA models such as StyleGAN2 (Karras et al., 2020b) and BigGAN (Brock et al., 2019) remain to be high, especially when trained with the image resolution on $1024 \times 1024$.

In this paper, our goal is to learn an unconditional GAN on high-resolution images, with low computational cost and few training samples. As summarized in Fig. 2, these training conditions expose the model to a high risk of overfitting and mode-collapse (Arjovsky & Bottou, 2017; Zhang & Khoreva, 2018). To train a GAN given the demanding training conditions, we need a generator ($G$) that can learn fast, and a discriminator ($D$) that can continuously provide useful signals to train $G$. To address these challenges, we summarize our contribution as:

- We design the Skip-Layer channel-wise Excitation (SLE) module, which leverages low-scale activations to revise the channel responses on high-scale feature-maps. SLE allows a more robust gradient flow throughout the model weights for faster training. It also leads to an automated learning of a style/content disentanglement like StyleGAN2.

- We propose a self-supervised discriminator $D$ trained as a feature-encoder with an extra decoder. We force $D$ to learn a more descriptive feature-map covering more regions from an input image, thus yielding more comprehensive signals to train $G$. We test multiple self-supervision strategies for $D$, among which we show that auto-encoding works the best.

- We build a computational-efficient GAN model based on the two proposed techniques, and show the model's robustness on multiple high-fidelity datasets, as demonstrated in Fig. 1.

## 2 RELATED WORKS

**Speed up the GAN training**: Speeding up the training of GAN has been approached from various perspectives. Ngxande et al. propose to reduce the computing time with depth-wise convolutions. Zhong et al. adjust the GAN objective into a min-max-min problem for a shorter optimization path. Sinha et al. suggest to prepare each batch of training samples via a core-set selection, leverage the better data preparation for a faster convergence. However, these methods only bring a limited improvement in training speed. Moreover, the synthesis quality is not advanced within the shortened training time.

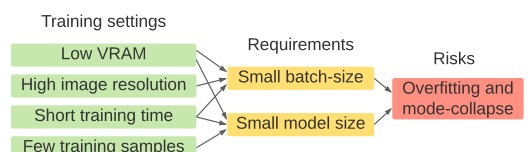

Figure 2: The causes and challenges for training GAN in our studied conditions.

**Train GAN on high resolution**: High-resolution training for GAN can be problematic. Firstly, the increased model parameters lead to a more rigid gradient flow to optimize $G$. Secondly, the target distribution formed by the images on $1024 \times 1024$ resolution is super sparse, making GAN much harder to converge. Denton et al. (2015); Zhang et al. (2017); Huang et al. (2017); Wang et al. (2018); Karras et al. (2019); Karnewar & Wang (2020); Karras et al. (2020b); Liu et al. (2021) develop the multi-scale GAN structures to alleviate the gradient flow issue, where $G$ outputs images and receives feedback from several resolutions simultaneously. However, all these approaches further increase the computational cost, consuming even more GPU memory and training time.

**Stabilize the GAN training**: Mode-collapse on $G$ is one of the big challenges when training GANs. And it becomes even more challenging given fewer training samples and a lower computational budget (a smaller batch-size). As $D$ is more likely to be overfitting on the datasets, thus unable to provide meaningful gradients to train $G$ (Gulrajani et al., 2017).

Prior works tackle the overfitting issue by seeking a good regularization for $D$, including different objectives (Arjovsky et al., 2017; Lim & Ye, 2017; Tran et al., 2017); regularizing the gradients (Gulrajani et al., 2017; Mescheder et al., 2018); normalizing the model weights (Miyato et al., 2018); and augmenting the training data (Karras et al., 2020a; Zhao et al., 2020). However, the effects of these methods degrade fast when the training batch-size is limited, since appropriate batch statistics can hardly be calculated for the regularization (normalization) over the training iterations.

Meanwhile, self-supervision on $D$ has been shown to be an effective method to stabilize the GAN training as studied in Tran et al. (2019); Chen et al. (2019). However, the auxiliary self-supervision tasks in prior works have limited using scenario and image domain. Moreover, prior works only studied on low resolution images ($32^2$ to $128^2$), and without a computing resource limitation.

## 3 METHOD

We adopt a minimalistic design for our model. In particular, we use a single conv-layer on each resolution in $G$, and apply only three (input and output) channels for the conv-layers on the high resolutions ($\geq 512 \times 512$) in both $G$ and $D$. Fig. 3 and Fig. 4 illustrate the model structure for our $G$ and $D$, with descriptions of the component layers and forward flow. These structure designs make our GAN much smaller than SOTA models and substantially faster to train. Meanwhile, our model remains robust on small datasets due to its compact size with the two proposed techniques.

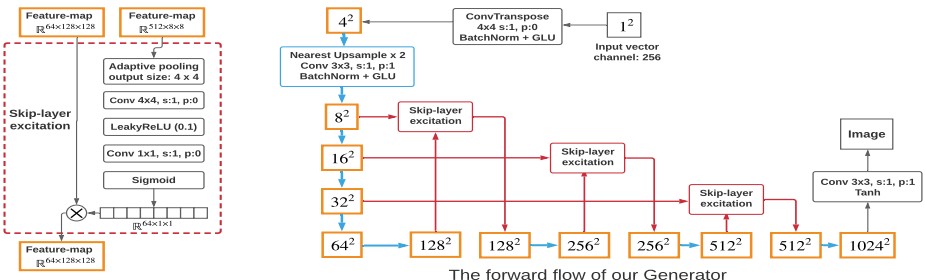

Figure 3: The structure of the skip-layer excitation module and the Generator. Yellow boxes represent feature-maps (we show the spatial size and omit the channel number), blue box and blue arrows represent the same up-sampling structure, red box contains the SLE module as illustrated on the left.

### 3.1 SKIP-LAYER CHANNEL-WISE EXCITATION

For synthesizing higher resolution images, the generator $G$ inevitably needs to become deeper, with more conv-layers, in concert with the up-sampling needs. A deeper model with more convolution layers leads to a longer training time of GAN, due to the increased number of model parameters and a weaker gradient flow through $G$ (Zhang et al., 2017; Karras et al., 2018; Karnewar & Wang, 2020). To better train a deep model, He et al. design the Residual structure (ResBlock), which uses a skip-layer connection to strengthen the gradient signals between layers. However, while ResBlock has been widely used in GAN literature (Wang et al., 2018; Karras et al., 2020b), it also increases the computation cost.

We reformulate the skip-connection idea with two unique designs into the *Skip-Layer Excitation* module (SLE). First, ResBlock implements skip-connection as an element-wise addition between the activations from different conv-layers. It requires the spatial dimensions of the activations to be the same. Instead of addition, we apply channel-wise multiplications between the activations, eliminating the heavy computation of convolution (since one side of the activations now has a spatial dimension of $1^2$). Second, in prior GAN works, skip-connections are only used within the same resolution. In contrast, we perform skip-connection between resolutions with a much longer range (e.g., $8^2$ and $128^2$, $16^2$ and $256^2$), since an equal spatial-dimension is no longer required. The two designs make SLE inherits the advantages of ResBlock with a shortcut gradient flow, meanwhile without an extra computation burden.

Formally, we define the Skip-Layer Excitation module as:

$$\mathbf{y} = \mathcal{F}(\mathbf{x}_{low}, \{\mathbf{W}_i\}) \cdot \mathbf{x}_{high} \tag{1}$$

Here $\mathbf{x}$ and $\mathbf{y}$ are the input and output feature-maps of the SLE module, the function $\mathcal{F}$ contains the operations on $\mathbf{x}_{low}$, and $\mathbf{W}_i$ indicates the module weights to be learned. The left panel in Fig. 3 shows an SLE module in practice, where $\mathbf{x}_{low}$ and $\mathbf{x}_{high}$ are the feature-maps at $8 \times 8$ and $128 \times 128$ resolution respectively. An adaptive average-pooling layer in $\mathcal{F}$ first down-samples $\mathbf{x}_{low}$ into $4 \times 4$

along the spatial-dimensions, then a conv-layer further down-samples it into $1 \times 1$. A LeakyReLU is used to model the non-linearity, and another conv-layer projects $\mathbf{x}_{low}$ to have the same channel size as $\mathbf{x}_{high}$. Finally, after a gating operation via a Sigmoid function, the output from $\mathcal{F}$ multiplies $\mathbf{x}_{high}$ along the channel dimension, yielding $\mathbf{y}$ with the same shape as $\mathbf{x}_{high}$.

SLE partially resembles the Squeeze-and-Excitation module (SE) proposed by Hu et al.. However, SE operates within one feature-map as a self-gating module. In comparison, SLE performs between feature-maps that are far away from each other. While SLE brings the benefit of channel-wise feature re-calibration just like SE, it also strengthens the whole model's gradient flow like ResBlock. The channel-wise multiplication in SLE also coincides with Instance Normalization (Ulyanov et al., 2016; Huang & Belongie, 2017), which is widely used in style-transfer. Similarly, we show that SLE enables $G$ to automatically disentangle the content and style attributes, just like StyleGAN (Karras et al., 2019). As SLE performs on high-resolution feature-maps, altering these feature-maps is shown to be more likely to change the style attributes of the generated image (Karras et al., 2019; Liu et al., 2021). By replacing $\mathbf{x}_{low}$ in SLE from another synthesized sample, our $G$ can generate an image with the content unchanged, but in the same style of the new replacing image.

### 3.2 SELF-SUPERVISED DISCRIMINATOR

Our approach to provide a strong regularization for $D$ is surprisingly simple. We treat $D$ as an encoder and train it with small decoders. Such auto-encoding training forces $D$ to extract image features that the decoders can give good reconstructions. The decoders are optimized together with $D$ on a simple reconstruction loss, which is only trained on real samples:

$$\mathcal{L}_{recons} = \mathbb{E}_{\mathbf{f} \sim D_{encode}(x), \, x \sim I_{real}}[||\mathcal{G}(\mathbf{f}) - \mathcal{T}(x)||], \tag{2}$$

where $\mathbf{f}$ is the intermediate feature-maps from $D$, the function $\mathcal{G}$ contains the processing on $\mathbf{f}$ and the decoder, and the function $\mathcal{T}$ represents the processing on sample $x$ from real images $I_{real}$.

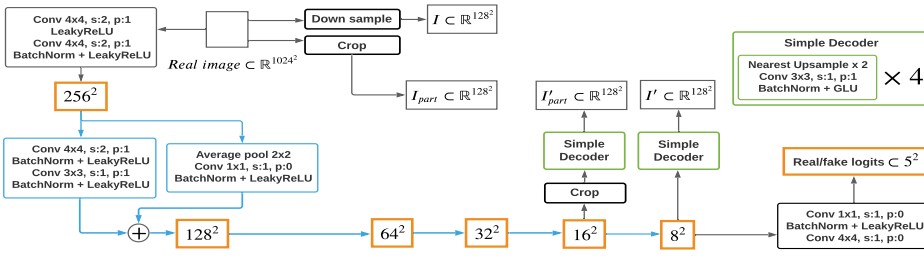

Figure 4: The structure and the forward flow of the Discriminator. Blue box and arrows represent the same residual down-sampling structure, green boxes mean the same decoder structure.

Our self-supervised $D$ is illustrated in Fig. 4, where we employ two decoders for the feature-maps on two scales: $\mathbf{f}_1$ on $16^2$ and $\mathbf{f}_2$ on $8^2$. The decoders only have four conv-layers to produce images at $128 \times 128$ resolution, causing little extra computations (much less than other regularization methods). We randomly crop $\mathbf{f}_1$ with $\frac{1}{8}$ of its height and width, then crop the real image on the same portion to get $I_{part}$. We resize the real image to get $I$. The decoders produce $I'_{part}$ from the cropped $\mathbf{f}_1$, and $I'$ from $\mathbf{f}_2$. Finally, $D$ and the decoders are trained together to minimize the loss in eq. 2, by matching $I'_{part}$ to $I_{part}$ and $I'$ to $I$.

Such reconstructive training makes sure that $D$ extracts a more comprehensive representation from the inputs, covering both the overall compositions (from $\mathbf{f}_2$) and detailed textures (from $\mathbf{f}_1$). Note that the processing in $\mathcal{G}$ and $\mathcal{T}$ are not limited to cropping; more operations remain to be explored for better performance. The auto-encoding approach we employ is a typical method for self-supervised learning, which has been well recognized to improve the model robustness and generalization ability (He et al., 2020; Hendrycks et al., 2019; Jing & Tian, 2020; Goyal et al., 2019). In the context of GAN, we find that a regularized $D$ via self-supervision training strategies significantly improves the synthesis quality on $G$, among which auto-encoding brings the most performance boost.

Although our self-supervision strategy for $D$ comes in the form of an auto-encoder (AE), this approach is fundamentally different from works trying to combine GAN and AE (Larsen et al., 2016;

Guo et al., 2019; Zhao et al., 2016; Berthelot et al., 2017). The latter works mostly train $G$ as a decoder on a learned latent space from $D$, or treat the adversarial training with $D$ as an supplementary loss besides AE's training. In contrast, our model is a pure GAN with a much simpler training schema. The auto-encoding training is only for regularizing $D$, where $G$ is not involved.

In sum, we employ the hinge version of the adversarial loss (Lim & Ye (2017); Tran et al. (2017)) to iteratively train our D and G. We find the different GAN losses make little performance difference, while hinge loss computes the fastest:

$$\mathcal{L}_D = - \mathbb{E}_{x \sim I_{real}}[min(0, -1 + D(x))] - \mathbb{E}_{\hat{x} \sim G(z)}[min(0, -1 - D(\hat{x})] + \mathcal{L}_{recons} \quad (3)$$
$$\mathcal{L}_G = - \mathbb{E}_{z \sim \mathcal{N}}[D(G(z))], \quad (4)$$

## 4 EXPERIMENT

**Datasets**: We conduct experiments on multiple datasets with a wide range of content categories. On $256 \times 256$ resolution, we test on Animal-Face Dog and Cat (Si & Zhu, 2011), 100-Shot-Obama, Panda, and Grumpy-cat (Zhao et al., 2020). On $1024 \times 1024$ resolution, we test on Flickr-Face-HQ (FFHQ) (Karras et al., 2019), Oxford-flowers (Nilsback & Zisserman, 2006), art paintings from WikiArt (wikiart.org), photographs on natural landscape from Unsplash (unsplash.com), Pokemon (pokemon.com), anime face, skull, and shell. These datasets are designed to cover images with different characteristics: photo realistic, graphic-illustration, and art-like images.

**Metrics**: We use two metrics to measure the models' synthesis performance: 1) Fréchet Inception Distance (FID) (Heusel et al., 2017) measures the overall semantic realism of the synthesized images. For datasets with less than 1000 images (most only have 100 images), we let $G$ generate 5000 images and compute FID between the synthesized images and the whole training set. 2) Learned perceptual similarity (LPIPS) (Zhang et al., 2018) provides a perceptual distance between two images. We use LPIPS to report the reconstruction quality when we perform latent space back-tracking on $G$ given real images, and measure the auto-encoding performance. We find it unnecessary to involve other metrics, as FID is unlikely to be inconsistent with the others, given the notable performance gap between our model and the compared ones. For all the testings, we train the models 5 times with random seeds, and report the highest scores. The relative error is less than five percent on average.

**Compared Models**: We compare our model with: 1) the state-of-the-art (SOTA) unconditional model, StyleGAN2, 2) a baseline model ablated from our proposed one. Note that we adopt Style-GAN2 with recent studies from (Karras et al., 2020a; Zhao et al., 2020), including the model configuration and differentiable data-augmentation, for the best training on few-sample datasets. Since StyleGAN2 requires much more computing-cost (cc) to train, we derive an extra baseline model. In sum, we compare our model with StyleGAN2 on the absolute image synthesis quality regardless of cc, and use the baseline model for the reference within a comparable cc range.

The baseline model is the strongest performer that we integrated from various GAN techniques based on DCGAN (Radford et al., 2015): 1) spectral-normalization (Miyato et al., 2018), 2) exponential-moving-average (Yazıcı et al., 2018) optimization on $G$, 3) differentiable-augmentation, 4) GLU (Dauphin et al., 2017) instead of ReLU in $G$. We build our model upon the baseline with the two proposed techniques: the skip-layer excitation module and the self-supervised discriminator.

Table 1: Computational cost comparison of the models.

|  |  | StyleGAN2@0.25 | StyleGAN2@0.5 | StyleGAN2 | Baseline | Ours |
|---|---|---|---|---|---|---|
| Resolution: $256^2$ Batch-size: 8 | Training time (hour / 10k iter) | 1 | 1.8 | 3.8 | 0.7 | 1 |
|  | Training vram (GB) | 7 | 16 | 18 | 5 | 6.5 |
|  | Model parameters (million) | 27.557 | 45.029 | 108.843 | 44.359 | 47.363 |
| Resolution: $1024^2$ Batch-size: 8 | Training time (hour / 10k iter) | 3.6 | 5 | 7 | 1.3 | 1.7 |
|  | Training vram (GB) | 12 | 23 | 36 | 9 | 10 |
|  | Model parameters (million) | 27.591 | 45.15 | 109.229 | 44.377 | 47.413 |

Table. 1 presents the normalized cc figures of the models on Nvidia's RTX 2080-Ti GPU, implemented using PyTorch (Paszke et al., 2017). Importantly, the slimed StyleGAN2 with $\frac{1}{4}$ parameters cannot converge on the tested datasets at $1024^2$ resolution. We compare to the StyleGAN2 with $\frac{1}{2}$ parameters (if not specifically mentioned) in the following experiments.

## 4.1 IMAGE SYNTHESIS PERFORMANCE

**Few-shot generation**: Collecting large-scale image datasets are expensive, or even impossible, for a certain character, a genre, or a topic. On those few-shot datasets, a data-efficient model becomes especially valuable for the image generation task. In Table. 2 and Table. 3, we show that our model not only achieves superior performance on the few-shot datasets, but also much more computational-efficient than the compared methods. We save the checkpoints every 10k iterations during training and report the best FID from the checkpoints (happens at least after 15 hours of training for Style-GAN2 on all datasets). Among the 12 datasets, our model performs the best on 10 of them.

Please note that, due to the VRAM requirement for StyleGAN2 when trained on $1024^2$ resolution, we have to train the models in Table. 3 on a RTX TITAN GPU. In practice, 2080-TI and TITAN share a similar performance, and our model runs the same time on both GPUs.

Table 2: FID comparison at $256^2$ resolution on few-sample datasets.

| | | | Animal Face - Dog | Animal Face - Cat | Obama | Panda | Grumpy-cat |
|---|---|---|---|---|---|---|---|
| | Image number | | 389 | 160 | 100 | 100 | 100 |
| Training time on one RTX 2080-Ti | 20 hour | StyleGAN2 | 58.85 | 42.44 | 46.87 | 12.06 | 27.08 |
| | | StyleGAN2 finetune | 61.03 | 46.07 | **35.75** | 14.5 | 29.34 |
| | 5 hour | Baseline | 108.19 | 150.3 | 62.74 | 15.4 | 42.13 |
| | | Baseline+Skip | 94.21 | 72.97 | 52.50 | 14.39 | 38.17 |
| | | Baseline+decode | 56.25 | 36.74 | 44.34 | 10.12 | 29.38 |
| | | Ours (B+Skip+decode) | **50.66** | **35.11** | 41.05 | **10.03** | **26.65** |

**Training from scratch vs. fine-tuning**: Fine-tuning from a pre-trained GAN (Mo et al., 2020; Noguchi & Harada, 2019; Wang et al., 2020) has been the go-to method for the image generation task on datasets with few samples. However, its performance highly depends on the semantic consistency between the new dataset and the available pre-trained model. According to Zhao et al., fine-tuning performs worse than training from scratch in most cases, when the content from the new dataset strays away from the original one. We confirm the limitation of current fine-tuning methods from Table. 2 and Table. 3, where we fine-tune StyleGAN2 trained on FFHQ use the Freeze-D method from Mo et al.. Among all the tested datasets, only Obama and Skull favor the fine-tuning method, making sense since the two sets share the most similar contents to FFHQ.

**Module ablation study**: We experiment with the two proposed modules in Table. 2, where both SLE (skip) and decoding-on-$D$ (decode) can separately boost the model performance. It shows that the two modules are orthogonal to each other in improving the model performance, and the self-supervised $D$ makes the biggest contribution. Importantly, the baseline model and StyleGAN2 diverge fast after the listed training time. In contrast, our model is less likely to mode collapse among the tested datasets. Unlike the baseline model which usually model-collapse after trained for 10 hours, our model maintains a good synthesis quality and won't collapse even after trained for 20 hours. We argue that it is the decoding regularization on $D$ that prevents the model from divergence.

Table 3: FID comparison at $1024^2$ resolution on few-sample datasets.

| | | | Art Paintings | FFHQ | Flower | Pokemon | Anime Face | Skull | Shell |
|---|---|---|---|---|---|---|---|---|---|
| | Image number | | 1000 | 1000 | 1000 | 800 | 120 | 100 | 60 |
| Training time on one RTX TITAN | 24 hour | StyleGAN2 | 74.56 | 25.66 | 45.23 | 190.23 | 152.73 | 127.98 | 241.37 |
| | | StyleGAN2 finetune | N/A | N/A | 36.72 | 60.12 | 61.23 | **107.68** | 220.45 |
| | 8 hour | Baseline | 62.27 | 38.35 | 42.25 | 67.86 | 101.23 | 186.45 | 202.32 |
| | | Ours | **45.08** | **24.45** | **25.66** | **57.19** | **59.38** | 130.05 | **155.47** |

Table 4: FID comparison at $1024^2$ resolution on datasets with more images.

| Model | Dataset | | Art Paintings | | | FFHQ | | | | Nature Photograph | | |
|---|---|---|---|---|---|---|---|---|---|---|---|---|
| | Image number | | 2k | 5k | 10k | 2k | 5k | 10k | 70k | 2k | 5k | 10k |
| StyleGAN2 | | | 70.02 | 48.36 | **41.23** | **18.38** | **10.45** | **7.86** | **4.4** | 67.12 | **41.47** | **39.05** |
| Baseline | | | 60.02 | 51.23 | 49.38 | 36.45 | 27.86 | 25.12 | 17.62 | 71.47 | 66.05 | 62.28 |
| Ours | | | **44.57** | **43.27** | 42.53 | 19.01 | 17.93 | 16.45 | 12.38 | **52.47** | 45.07 | 43.65 |

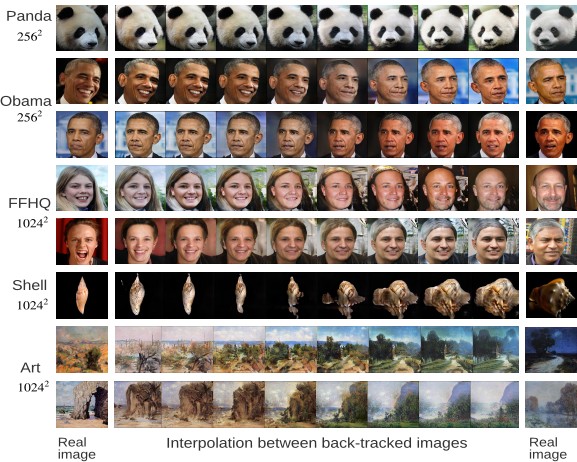

Figure 6: Latent space back-tracking and interpolation.

Table 5: LPIPS of back-tracking with $G$

|  | Cat | Dog | FFHQ | Art |
|---|---|---|---|---|
| Resolution | 256 | | 1024 | |
| Baseline @ 20k iter | 2.113 | 2.073 | 2.589 | 2.916 |
| Baseline @ 40k iter | 2.513 | 2.171 | 2.583 | 2.812 |
| Ours @ 40k iter | **1.821** | **1.918** | 2.425 | 2.624 |
| Ours @ 80k iter | 1.897 | 1.986 | **2.342** | **2.601** |

Table 6: FID of self-supervisions for $D$

|  | Art paintings | Nature photos |
|---|---|---|
| a. contrastive loss | 47.14 | 57.04 |
| b. predict aspect ratio | 49.21 | 59.22 |
| c. auto-encoding | **42.53** | **43.65** |
| d. a+b | 46.02 | 54.23 |
| e. a+b+c | 44.21 | 47.65 |

**Training with more images**: For more thorough evaluation, we also test our model on datasets with more sufficient training samples, as shown in Table. 4. We train the full StyleGAN2 for around five days on the Art and Photograph dataset with a batch-size of 16 on two TITAN RTX GPUs, and use the latest official figures on FFHQ from Zhao et al.. Instead, we train our model for only 24 hours, with a batch-size of 8 on a single 2080-Ti GPU. Specifically, for FFHQ with all 70000 images, we train our model with a larger batch-size of 32, to reflect an optimal performance of our model.

In this test, we follow the common practice of computing FID by generating 50k images and use the whole training set as the reference distribution. Note that StyleGAN2 has more than double the parameters compared to our model, and trained with a much larger batch-size on FFHQ. These factors contribute to its better performances when given enough training samples and computing power. Meanwhile, our model keeps up well with StyleGAN2 across all testings with a considerably lower computing budget, showing a compelling performance even on larger-scale datasets, and a consistent performance boost over the baseline model.

**Qualitative results**: The advantage of our model becomes more clear from the qualitative comparisons in Fig. 5. Given the same batch-size and training time, StyleGAN2 either converges slower or suffers from mode collapse. In contrast, our model consistently generates satisfactory images. Note that the best results from our model on Flower, Shell, and Pokemon only take three hours' training, and for the rest three datasets, the best performance is achieved at training for eight hours. For StyleGAN2 on "shell", "anime face", and "Pokemon", the images shown in Fig. 5 are already from the best epoch, which they match the scores in Table. 2 and Table. 3. For the rest of the datasets, the quality increase from StyleGAN2 is also limited given more training time.

## 4.2 MORE ANALYSIS AND APPLICATIONS

**Testing mode collapse with back-tracking**: From a well trained GAN, one can take a real image and invert it back to a vector in the latent space of $G$, thus editing the image's content by altering the back-tracked vector. Despite the various back-tracking methods (Zhu et al., 2016; Lipton & Tripathi, 2017; Zhu et al., 2020; Abdal et al., 2019), a well generalized $G$ is arguably as important for the good inversions. To this end, we show that our model, although trained on limited image samples, still gets a desirable performance on real image back-tracking.

In Table 5, we split the images from each dataset with a training/testing ratio of 9:1, and train $G$ on the training set. We compute a reconstruction error between all the images from the testing set and their inversions from $G$, after the same update of 1000 iterations on the latent vectors (to prevent the vectors from being far off the normal distribution). The baseline model's performance is getting worse with more training iterations, which reflects mode-collapse on $G$. In contrast, our model gives better reconstructions with consistent performance over more training iterations. Fig. 6 presents the back-tracked examples (left-most and right-most samples in the middle panel) given the real images.

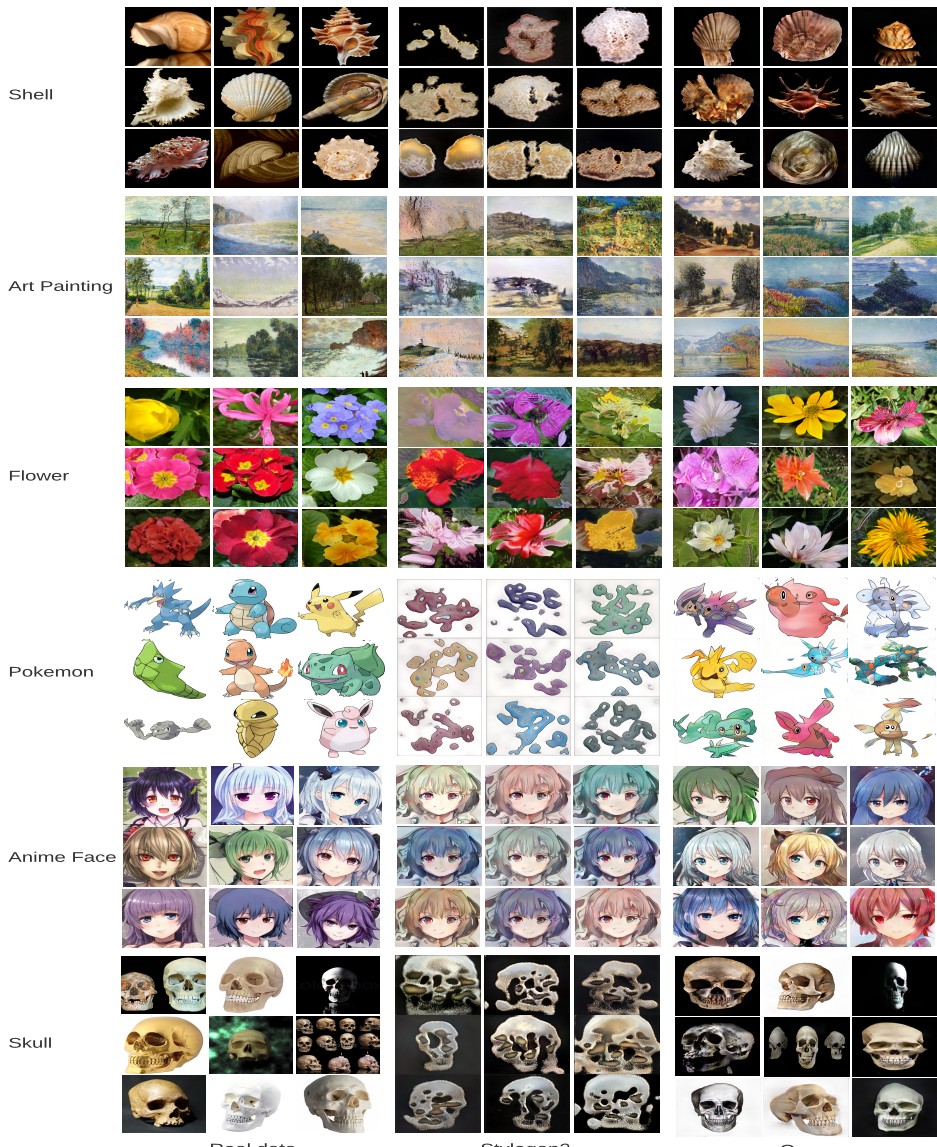

Figure 5: **Qualitative comparison between our model and StyleGAN2** on $1024^2$ resolution datasets. The left-most panel shows the training images, and the right two panels show the un-curated samples from StyleGAN2 and our model. Both models are trained from scratch for 10 hours with a batch-size of 8. The samples are generated from the checkpoint with the lowest FID.

The smooth interpolations from the back-tracked latent vectors also suggest little mode-collapse of our $G$ (Radford et al., 2015; Zhao et al., 2020; Robb et al., 2020).

In addition, we show qualitative comparisons in appendix D, where our model maintains a good generation while StyleGAN2 and baseline are model-collapsed.

**The self-supervision methods and generalization ability on** $D$: Apart from the auto-encoding training for $D$, we show that $D$ with other common self-supervising strategies also boost GAN's performance in our training settings. We test five self-supervision settings, as shown in Table 6, which all brings a substantial performance boost compared to the baseline model. Specifically, setting-a refers to contrastive learning which we treat each real image as a unique class and let $D$ classify them. For setting-b, we train $D$ to predict the real image's original aspect-ratio since they are reshaped to square when fed to $D$. Setting-c is the method we employ in our model, which

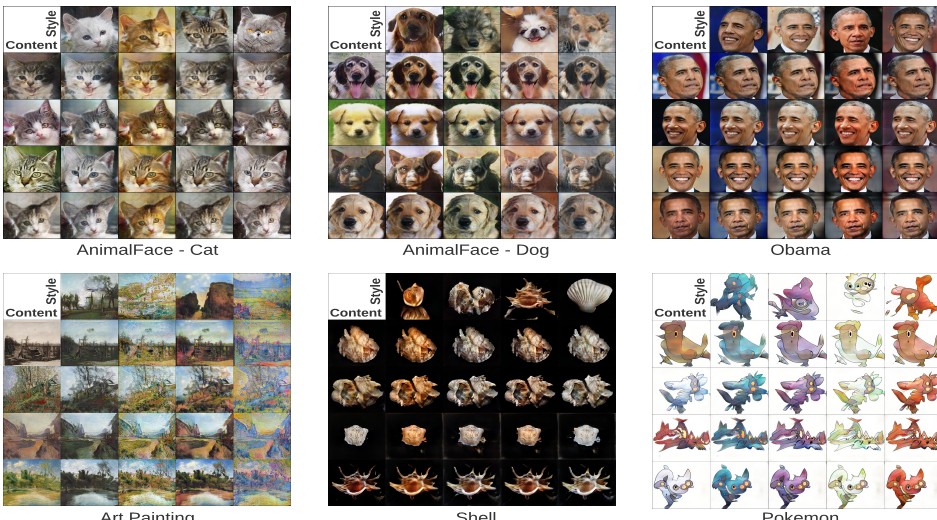

AnimalFace - Cat     AnimalFace - Dog     Obama

Art Painting     Shell     Pokemon

Figure 7: **Style-mixing results** from our model trained for only 5 hours on single GPU.

trains $D$ as an encoder with a decoder to reconstruct real images. To better validate the benefit of self-supervision on $D$, all the testings are conducted on full training sets with 10000 images, with a batch-size of 8 to be consistent with Table 4. We also tried training with a larger batch-size of 16, which the results are consistent to the batch-size of 8.

Interestingly, according to Table 6, while setting-c performs the best, combining it with the rest two settings lead to a clear performance downgrade. The similar behavior can be found on some other self-supervision settings, e.g. when follow Chen et al. (2019) with a "rotation-predicting" task on art-paintings and FFHQ datasets, we observe a performance downgrade even compared to the baseline model. We hypothesis the reason being that the auto-encoding forces $D$ to pay attention to more areas of the input image, thus extracts a more comprehensive feature-map to describe the input image (for a good reconstruction). In contrast, a classification task does not guarantee $D$ to cover the whole image. Instead, the task drives $D$ to only focus on small regions because the model can find class cues from small regions of the images. Focusing on limited regions (i.e., react to limited image patterns) is a typical overfitting behavior, which is also widely happening for $D$ in vanilla GANs. More discussion can be found in appendix B.

**Style mixing like StyleGAN**. With the channel-wise excitation module, our model gets the same functionality as StyleGAN: it learns to disentangle the images' high-level semantic attributes (style and content) in an unsupervised way, from $G$'s conv-layers at different scales. The style-mixing results are displayed in Fig. 7, where the top three datasets are $256 \times 256$ resolution, and the bottom three are $1024 \times 1024$ resolution. While StyleGAN2 suffers from converging on the bottom high-resolution datasets, our model successfully learns the style representations along the channel dimension on the "excited" layers (i.e., for feature-maps on $256 \times 256$, $512 \times 512$ resolution). Please refer to appendix A and C for more information on SLE and style-mixing.

## 5 CONCLUSION

We introduce two techniques that stabilize the GAN training with an improved synthesis quality, given sub-hundred high-fidelity images and a limited computing resource. On thirteen datasets with a diverse content variation, we show that a skip-layer channel-wise excitation mechanism (SLE) and a self-supervised regularization on the discriminator significantly boost the synthesis performance of GAN. Both proposed techniques require minor changes to a vanilla GAN, enhancing GAN's practicality with a desirable plug-and-play property. We hope this work can benefit downstream tasks of GAN and provide new study perspectives for future research.

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
