# OpenReview forum: "Towards Faster and Stabilized GAN Training for High-fidelity Few-shot Image Synthesis"
_ICLR.cc/2021/Conference — ICLR 2021 Poster_

### Official Review · AnonReviewer3 · 2020-10-23
**Interesting work but lacks polish and proper comparisons**

**Rating:** 7
**Confidence:** 5

**Review:**

In this paper, the authors introduce a new framework for unconditional image generation. The introduce a skip-layer excitation module (SLE) that allows gradient flow between activations of different spatial size. They also included a discriminator that is forced to autoencode the image. The authors claim that their framework is able to produce images of higher quality compared to SOTA with less resources.

1) Figures and tables are not great
Figure 3: The figure on right flows from top to bottom to top, making it a little difficult to follow. Arrows sometimes correspond to an operation (upsampling) or the workflow (the red arrows). Different fonts are used.

Figure 4: Similarly, hard to follow. Arrows aren’t straight. What’s the resolution of the crop?

Figure 5: StyleGAN2 and proposed method’s image sizes should be the same for easy qualitative comparison. Perhaps plot real images in a single row at the top. Current organization is difficult to follow.

Table 2:organization can be better. What is stylegan2 ft?

2) The choice of sigmoid sigmoid in SLE is not justified, especially since AdaIN[1], a very similar method, does not use sigmoid.

3) No intuition is given for cropping the feature maps for decoder. Why does it help?

4) Experiments in table 2 and table 3 are computed with different gpus which makes their results/training time incomparable.

5) Similarly, comparisons with StyleGAN2 have different training time and batch sizes. I understand the point the authors are trying to point out is with less resources, their model performs better. However, it is unclear if their model would mode collapse with longer training or would it perform worse on bigger batches. I would like to see a fair comparison and also a line plot of FID w.r.t training time.

6) Under 4.2, the authors claim that a well generalized G is the key for good inversions. This claim is not true since Image2StyleGAN[2] showed that even a network with randomly initialized weights can achieve good inversion.

7) Under 4.2 the authors claim that “The smooth interpolations from the back-tracked latent vectors also suggest little overfitting and mode-collapse of our G.” I am aware that in StyleGAN[3], they claim smoothness of interpolation in latent space is correlated with how disentangled the latent space. But I am not aware of work that show interpolation is related to overfitting and mode collapse. Citation is needed.

8) How good are the decoders? A figure showing the reconstruction will be good.

9) For comparisons with other self-supervision method, the experimental setup is missing.

10) In which layers are style mixing being performed on? In my opinion, the style mixing results show more of a color change rather than anything semantically meaningful. StyleGAN showed that styles from different layers correspond to different semantic concepts. This seems to be missing in this model.


General comments: Writing can be improved. Some sentences can be rewritten to flow better
1)	“The biggest challenge lies on the overfitting issue on D, thus leading to mode-collapse on G, given the sparse training samples and a low computing budget.”
2)	“elevated model depth”
3)	“In contrast, our model maintains a good synthesis quality, even double the training time, thanks to the decoding regularization of D”
4)	“at most an eight hours’ training is needed”

Overall, I think the idea of SLE is interesting, but clear comparisons and ablations should be done to validate its usefulness.

[1] Huang, Xun, and Serge Belongie. "Arbitrary style transfer in real-time with adaptive instance normalization." Proceedings of the IEEE International Conference on Computer Vision. 2017.

[2] Abdal, Rameen, Yipeng Qin, and Peter Wonka. "Image2StyleGAN: How to Embed Images Into the StyleGAN Latent Space?." (2019).

[3] Karras, Tero, Samuli Laine, and Timo Aila. "A style-based generator architecture for generative adversarial networks." Proceedings of the IEEE conference on computer vision and pattern recognition. 2019.

---

> ### Author Response · Authors · 2020-11-11
> **Response to concern 1 to 5**
>
> To reviewer #3:
>
> Firstly, We really appreciate the reviewer's detailed review, and we know that you are taking your valuable time and effort to help our paper. We are respectful and grateful to the reviewer in our response. We hope you don't mind our long response, and looking forward to hearing your response.
>
> 1.1. Figure 3: We have changed to the same fonts.
> All the arrows representing a module in the network, so they actually are consistent. Blue arrow: the upsampling conv module; red arrow: the SLE module; grey arrow: the basic conv module.
>
> 1.2. Figure 4: We have straightened the arrows.
> The cropping is a design choice that is not fixed, so we do not specify the cropping size. Please note that the self-supervised discriminator is a general framework that is well abstracted. Any detailed implementation (such as cropping size) can be different.
>
> 1.3. Figure 5: We have updated the figure given more page space. Please check.
>
> 1.4. Table 2: StyleGAN-ft means we fine-tune the styleGAN from a pre-trained model on the FFHQ dataset. It has been stated in the paper.
>
> 2. The detailed implementation of SLE is not fixed. It can vary on different datasets. Sigmoid is used in SENet, and we generally find SLE with sigmoid performs better on all tested datasets. Please note that, in AdaIN, the channel-wise multiplication is performed between the content feature-map and the style feature-map, and the feature-maps are batch-normalized. So the values used to multiply the content feature-map are actually well restricted within a small value range around 0. Sigmoid can well simulate such behavior and restrict the values in a similar range, therefore help avoid some imbalanced peak weights.
>
> 3. Cropping is not for better performance, but lower computation cost. Cropping makes sure the image reconstruction can be covered on the full image at 1024 resolution while also making sure to reconstruct the detailed textures that cannot be represented in a 128 resolution full image. If without cropping and doing a naive image reconstruction on 1024 resolution, the computing cost is much higher (4^3 more convolution operations if doing reconstruction on 1024 resolution than 128 resolution).
>
> 4. The results are comparable because the GPUs (TITAN RTX and 2080TI) share almost identical performance in practice. Their major difference is that TITAN has more VRAMS, which is needed to train StyleGAN2 (with the same batch size, StyleGAN2 takes much more VRAM than our model). This is the only reason we do table 2 on TITAN. As we already tested, TITAN and 2080TI takes the same time to train our model and baseline. So, please feel free to compare between table 2 and 3.
>
> 5. We follow the common practice of reporting “the best-performed iteration during the training time” for the models, as stated in the paper sec 4.1, so a model cannot get a worse score due to longer training time. Table 4 shows that our model performs better on a larger batch size. It is unlikely that a GAN performs worse with a larger batch size than 8. And we do not understand the reviewer’s question ”if their model would mode collapse with longer training.” To our knowledge and experience, all the GANs will collapse, given longer training time. We argue that the experiments in Tables 2 and 3 are already fair, with the same model size (weights number) and the same batch size. This paper is set on a low computation budget, and our methods are proposed given computation-efficient requirements; experiments with more computing costs goes against our goal.

---

> > ### Author Response · Authors · 2020-11-11
> > **Response to concern 6 to 10**
> >
> > 6. We cannot agree with the reviewer on this point. Firstly, in Image2StyleGAN[2], the inversion from a model with random weights is clearly worse than from a trained model; see Fig 5 (b)(e) and Fig 14 (b)(e). Secondly, a randomly initialized model is arguably more “generalized” than a trained model, because training is the process of restricting the model to fit into a specified dataset. We also show such observation in our appendix B table 1, where a random weight Discriminator performs better in feature-extraction compared to a trained Discriminator on baseline model.
> > We are using the inversion performance on unseen images to determine if the generator is well generalized on the trained domain. We think our statement holds true and does not conflict with the reviewer’s reference. Please consider that, if a G is not generalized after training, it means it can only generate a few images regardless of the input latent vector. Thus it cannot invert any unseen images. In contrast, if a generator can invert many more images, it means it is more generalized (so does a randomly initialized model, which is generalized well and able to invert more images). A random weight model can be generalized but cannot simulate the training distribution (and gets low FID). In contrast, our model is both well-generalized and having a better FID.
> >
> > 7.
> > There are plenty of works using a smooth interpolation to show if the model is overfitting. For some latest paper references, please see Fig 4 in Robb, et al. and Fig 7 in Zhao, et al.
> > The reasoning behind it is pretty intuitive. If a model is mode collapse, it can only generate a few images. Its interpolation cannot be smooth but instead discontinue, probably with half from the left mode and half from the right mode. In contrast, if a model is not mode collapse, it can generate many more images; thus, an interpolation between any two images is smooth with meaningful and slightly different images in between.
> > Robb, Esther, et al. "Few-Shot Adaptation of Generative Adversarial Networks." arXiv preprint arXiv:2010.11943 (2020).
> > Zhao, Shengyu, et al. "Differentiable augmentation for data-efficient gan training." Advances in Neural Information Processing Systems 33 (2020).
> >
> >
> > 8. The decoders are pretty good. We have updated appendix sec E for a qualitative result from decoder, please check. Their reconstruction performance is high. On average, a 0.5 LPIPS score is reached for all the tested datasets. The code we provided will also save the decoder reconstruction images during training and also report the LPIPS of the decoder.
> >
> > 9. Thanks for the reminder. We have updated the paper with the setup in sec 4.2.
> >
> > 10.  Style mixing in our model is performed via the SLE modules, so it is only performed on the higher resolutions (128, 256, 512). Please check appendix C for more information on style-mixing. The semantics usually lies on the lower layers (8,16,32), which is why stylegan can learn them, and our model can only transfer the colors. As shown in StyleGAN (Karras et al.) Fig 3 bottom rows, it also can only transfer colors on high-resolution layers.
> > Moreover, since StyleGAN can hardly converge in the few-shot datasets, we can not properly compare them given the big overall image generation quality difference. We do not intend to compete on style-mixing also.
> >
> > Karras, Tero, Samuli Laine, and Timo Aila. "A style-based generator architecture for generative adversarial networks." Proceedings of the IEEE conference on computer vision and pattern recognition. 2019.
> >
> > General comments:
> > We appreciate the detailed help from the reviewer, we have updated our paper with revised sentences and presentations.
> > Please also check appendix section A for a detailed experiment on SLE, in which we show the performance boost from SLE.

---

> > > ### Comment · AnonReviewer3 · 2020-11-20
> > > **Response**
> > >
> > > 1.4.4 Even if they have similar performances, I think it will still be more fair to run everything on the same GPU.
> > >
> > > 1.4.6 I am not sure "if a G is not generalized after training, it means it can only generate a few images regardless of the input latent vector." is always true. Assume that G mode collapses and is only able to generate a few images regardless of the input vector sampled from normal distribution, would it not be able to generate other non-repeated images when given an input latent vector NOT sampled from the normal distribution? And even if the claims are true, I am not certain that it is fair to make this claim without evidence. There are no work cited in the paper with any theoretical and empirical results justifying this claim.
> > >
> > > Otherwise, I am satisfied with the clarification from the authors. I will be revising my recommendation to an accept,

---

> > > > ### Author Response · Authors · 2020-11-20
> > > > **Thank you for your response**
> > > >
> > > > Thank you for your suggestions and discussions.
> > > >
> > > > 1. We will follow your suggestion to include the training time of different models on the same GPU, and adding more description for Table2 and 3, to make it easier for the readers to compare with.
> > > >
> > > > 2. We agree with the reviewer's point that the generalization measurement is not strong currently. And we believe the reviewer makes a good point that "the back-tracked latent" should not be far off from the normal distribution. We will report the back-tracked latent vectors and show how far away they are from the mean of the normal distribution, and also the probability of sampling these vectors from the normal distribution.  We believe this will make Table.5 more solid.
> > > >
> > > > And we will make clear the detailed back-tracking setting in the paper for Table. 5: on both our model and baseline model, we update the vectors for the same 1000 iterations with the same learning rate, which constraints the vectors' value range to not be too far off the normal distribution. In other words, by updating for the same iterations, our model can find a better latent vector to reconstruct a real image.
> > > >
> > > > We will also cite more references on this point.
> > > >
> > > >
> > > > We appreciate the reviewer's positive feedback, and thanks again for your help!

---

### Official Review · AnonReviewer2 · 2020-10-27
**Borderline**

**Rating:** 7
**Confidence:** 4

**Review:**

Paper summary

This work studies training GANs on small datasets (in a few-shot setting) for high-resolution image synthesis. To generate high-quality samples with minimum computation cost, and to alleviate overfitting and training instabilities, two techniques are proposed: 1) For the generator the Skip-Layer channel-wise Excitation (SLE) module is introduced, “skip-connecting” low-scale layers with high-scale ones, which facilitates the gradient flow and allows style-content mixing from different images. 2) The discriminator is trained with additional small decoders to reconstruct given images, which acts as self-supervision and helps to reduce overfitting. Experiments show that the proposed GAN model copes well with high-resolution image synthesis task (256x256 – 1024x1024) while being trained on small datasets (down to 60 – 100 images), providing a significant speed up for this setting compared to existing approaches.

Strengths

1) The paper proposes an approach for a very challenging and important task of training GANs with the small amount of training data. To my knowledge, this paper is the first to generate high-resolution realistic images from datasets of such a small scale. This is valuable, as it potentially extends the domain of possible GAN applications. It is also good to see that the paper compares to the recent advances in low-data GANs (Karras et al., 2020a).

2) The paper achieves good results. The performance gain in comparison to StyleGAN2 and the considered baseline is visible across multiple small-scale datasets, see Tables 2 and 3. The improvement in visual quality is also clearly seen from Figure 5, though the evaluation setting might be unfair to StyleGAN2 (see my comments below).

3) SLE module seems interesting, as it is a novel way for designing a skip-connection between layers of different spatial resolutions in the generator. Besides facilitating the gradient flow, which helps the generator to learn quicker, it serves as a tool for style-content mixing by modulating high-scale features on low-scale encoding of another image.


Weaknesses

1) Limited technical novelty and missing comparisons to the closely related work. Each of the two proposed technical solutions have been proposed in similar forms in previous works, and the paper does not directly compare with them.

- SLE:
The proposed SLE module is a combination of skip connection + plus channel attention mechanism. However, no clear comparison with the related work is provided. As skip connections, one could also use simply residual connections, or use MSG-GAN (Karnevar & Wang, 2019) or StyleGAN2 approaches to improve the gradient flow (see Fig. 7 in the StyleGAN2 paper). They are probably heavier in terms of training speed, but I think there has to be an ablation where the proposed SLE is fairly compared to other ways of using skip connections to improve the gradient flow (memory, speed, performance).
From channel attention mechanism, the proposed technique resembles the Squeeze-and-Excitation module (SE) proposed by Hu et al. However, there were other follow up works that show superior results, such as Efficient Channel Attention (ECA) module proposed by Wang et al. CVPR’2020 or Convolutional Block Attention Module by Woo et al. ECCV’18. The paper doesn’t compare the proposed SLE to the above methods, thus it’s hard to judge how effective it is in comparison.

- SS-discriminator:
The employed auto-encoding approach is a typical method for self-supervised learning. However, there has been a line of works which uses different self-supervision techniques (e.g. Auxiliary Rotation Loss in Chen et al. CVPR 2019) or regularizations on the discriminator side (e.g. Zhao et al. 2020) for the same purpose as the proposed self-supervision, and it would be beneficial to see the comparison of the proposed s-s to existing approaches. Table 6 provides only the comparison with the two SS techniques, which might be suboptimal for the task at hand.

Generally, the proposed model compares well to the considered baseline (DCGAN + extras), but the need for the proposed solutions is not totally justified. Other similar existing solutions, mentioned above, implemented on top of the baseline, potentially could lead to the same performance improvement.

2) Incomplete evaluation and lack of the experimental support of some claims.

-	The comparison is done using only one metric - FID. This metric is known for not being able to detect overfitting, and, as was recently shown, is not a proper metric in low-data regimes (see Fig.4 in [*]). This raises the concern that on such small datasets the metric simply shows the degree of overfitting to the training set. With a limited training time used for the reported experiments this is naturally simpler to achieve for the proposed (lighter) model than for StyleGAN2, which might explain such a performance gap between two models. Overall, I disagree with the claim “We find it unnecessary to involve other metrics”, as the used FID metric could be misleading. It would be beneficial to employ also other metrics to measure the diversity of the generated samples. Thus, I find the evaluation presented in Tables 2,3 incomplete.

[*] Robb et al. FEW-SHOT ADAPTATION OF GENERATIVE ADVERSARIAL NETWORKS, ArXiv’2020.

- Overfitting is certainly one of the main challenges in a few-shot synthesis setting. However, the paper pays relatively small attention on analyzing the issue. Also in its current state it’s not clear if the proposed solutions actually help to avoid overfitting. On small datasets, the generated images probably resemble the training examples. This is seen for “Skull” in Figure 5, where for each generated image one could find the similar training example, also noticeable for “Shell” in Figure 6, where the interpolations tend to resemble the image on the left or on the right.
I agree that Table 5 is valuable, and that it shows relative overfitting strength compared to the baseline. However, I would also expect the analysis on the absolute values, as well as the comparison to StyleGAN2. For example, reporting LPIPS to the nearest training example would be helpful, together with showing generated samples together with the closest training examples.

-	Looking at the results in Table 4, as the number of training images become larger, the StyleGAN2 outperforms the proposed model. This also shows that even with a half of the parameters the StyleGAN2 model capacity is too large for less than 2k images. So I would expect for a low data regime setting StyleGAN2 with fewer parameters may have better results than the halved StyleGAN2. Given also that on larger datasets (5k-10k, Table 4) the proposed model underperforms, the problem might be the limited capacity of the proposed model, so increasing the number of parameters, e.g. number of channels, might help to improve the overall performance.  This trade-off between the model capacity and the size of the training set is not analysed in the paper, and from my point of view lead to unfair comparison in Tables 2,3,4.

-	For Figure 5, the images shown are from a different epoch than in Tables 3 and 4. Moreover, it might be unfair to clip StyleGAN2 at 10 hours of training, as its best epochs are coming later. The figure illustrates the speed-up from the proposed model, but the reader cannot match it to FID values in tables, also it is not possible to see the performance of StyleGAN2 in its best checkpoints on the studied settings.

-	It is unclear, what is meant by the “robustness” of the model in the paper. The model is claimed to be robust, but this claim is not really explained and supported experimentally.


Minor

1) Why also not to employ the SLE module in the discriminator?
2) How does the style mixing via SLE compares to other approaches, e.g. StyleGAN2?

Overall, I give the paper a borderline rating. I note that the paper studies an important problem, achieves good results, and advances GANs extending their application areas. On the other hand, I find some incompleteness in the experimental evaluation and unsupported claims in the paper, and have concerns about the limited novelty of the proposed technical solutions.

Post-rebuttal:
I believe the authors have done sufficient work in their revision to address my concerns. Thus I'm leaning towards acceptance and raising my score to a 7.

---

> ### Author Response · Authors · 2020-11-11
> **Response on reviewer's concern 1: novelty**
>
> First of all, we want to thank the reviewer for your efforts and help. We definitely appreciate your review and have made changes in the paper to better address your concerns. In our rebuttal, although there are few points of disagreement that we hope the reviewer could kindly have a look at, we are definitely respectful and grateful to the reviewer.
>
> 1.1
> We would like to emphasize that our novelty lies in this “combination” instead of the separated “Skip Connection (SC)” or “Channel Attention (CA)”. SC and CA were proposed individually for different purposes with different motivations (SC for gradient flow; CA for recalibrating channel-wise feature), and CA is rarely studied in GANs. Meanwhile, with yet another different motivation of “maintaining good performance while making the model lightweight”, we point out that SC and CA can be combined smoothly, yielding both great performance and computing efficiency. We argue that it is not simple to come up with the idea of combining SC and CA, and no prior work has done it to our knowledge. Just like the reviewer cited, prior works dive deeper towards either SC or CA, yet these two methods rarely show up together, nor to say that SC and CA can be combined. This paper is the first one that brings SC and CA together, and more importantly, shows that SC and CA cooperate well to solve our task: CA helps reduce the computational burden brought from SC, and SC helps CA to work on long-ranged different layers in the network.
>
> With that being said, we would like the reviewer to reconsider the necessity of comparing SLE to the “related work” that the reviewer proposed. SLE is a mechanism of combining SC and CA, not a fixed structure. In other words, more advanced structures towards SC and CA are inclusive in SLE. These advanced techniques such as (ECA, Wang, et al, CBA, Woo, et al)  are just a part of SLE (or to say, one of the design choices of SLE), rather than counterparts and competitors to SLE.
>
> 1.2
> We are proposing “SS-discriminator”: using self-supervision (SS) as a regularization for the discriminator, instead of just proposing an “auto-encoding” method. The idea is about integrating “self-supervision” to GAN, which no prior work has dedicated this idea to our knowledge (at the start of this work). And we show in table.6 that all SS brings benefit to GAN training, which auto-encoding works the best. The reviewer provided references (Chen et al, 2019, Zhao et al. 2020) are good instances of SS methods that can also boost GAN performance. Again, they are inclusive of our idea of SS-discriminator, rather than counterparts and competitors. More specifically, Chen's method is limited by the image domain to be rotation-sensitive, datasets like shells, blanket patterns, anime faces, are rotation-invariant and cannot use this method. In contrast, auto-encoding is more general that can be applied to more image domains. We will add them to the reference, however, we don't think a comparison is needed.  Btw, the reviewer did not provide which paper is for (Zhao et al. 2020).

---

> > ### Author Response · Authors · 2020-11-11
> > **Response on reviewer's concern 2: experiments and claims**
> >
> > Please check our updated Appendix sec D. We have two figures showing the performance difference between our model, the baseline, and StyleGAN2, where the baseline model and StyleGAN2 all suffer from mode collapse. In contrast, our model can generate diverse images.
> >
> > 2.1
> > Please note that FID is reliable in reporting the relative image quality and is sensitive to mode collapse. If one model has a clearly worse (>20%) FID score than another, it is unlikely that it is better.
> >
> > Importantly, in table 2 and 3, the FID difference between our model and the StyleGAN2 is usually larger than 20%, and the FID on several datasets for the baseline models means that it is mode collapsed or not even able to converge. Given such a big difference, it indicates a clear visual quality difference. The reviewer cited paper, which (Robb et al.) leverages the other metrics, notes that the tested models' FID difference is quite small (less than 5% relative difference).
> >
> > Moreover, Robb et al. show the case with only 10 images in the training set, which does not apply well in our case. Instead, we study the sub-hundreds image in the training set.  A latent space defined by 100 data points can be exponentially complex than 10 data points. Thus their case is incomparable to our case. Besides, Fig 4 from (Robb et al.) shows the discontinuous interpolation and has defects in between. In contrast, in our paper, Fig 6 shows that our interpolation continues well without defects, even for back-tracked images.
> >
> > Please also refer to Figure 5 and Figure 4 in the appendix for a clear qualitative difference. These factors summarize why we say “it unnecessary to involve other metrics.” Importantly, our model performs better, even with much less computing power (Tables 2 and 3 are already a very unfair setting to our model). We hope the reviewer don't worry about our model's claimed performance.
> >
> > 2.2:
> > The reviewer raised a good point on “overfitting” on GANs. However, we argue that such an “overfitting” concern is not a well-defined topic yet and beyond this paper's scope. For example, GAN is a distribution-simulation algorithm, given 100 images as a training set. If a generator can perfectly generate all the 100 images, should we say it is bad and “overfitted,” or should we say it is good and well done in simulating the distribution?
> >
> > As the reviewer asked the LPIPS score, however, a low LPIPS can both mean that the generated images are diverse and high quality (each generated image looks similar to a different real image) and can also mean that all generated samples are collapsed to the same real sample. And a higher LIPIS can mean all the generated images are bad quality, or good quality but unlike any real sample. We find it hard to make sense of this LPIPS score.
> >
> > Therefore, we argue that the term “overfitting” is currently hard to quantify for considering a Generator's performance. Instead, qualitative results are more intuitive to check. The “generalization ability” of a generator can be reflected through sample interpolations, shown in Fig 6; mode collapse and overall image quality are reflected from FID, shown in Table 2,3,4, and Fig 5,6,7, Appendix Fig 3.
> >
> > Meanwhile, we agree that the “overfitting” concern is definitely worth future study but stray away from this paper’s focus.
> >
> > With all been said, we hope the reviewer could reconsider our paper on the evaluation part.
> >
> >
> > 2.3:
> > Please check our response to reviewer 4 - 1 and to reviewer 1 - 3.  Table.4 shows the full-size StyleGAN2, not a 0.5 sized one.
> > Our goal is to find a smaller model that is computationally efficient and works on a small training set. Therefore, experiments that increase our model’s capacity go the opposite of our goal. If one has limited resources and fewer training samples, our model is favored over a same-sized stylegan. The “trade-off” between model size and training set size is a more general study topic that is beyond the scope of this paper, especially given the limited pages.
> >
> >
> > 2.4:
> > Actually, there is no inconsistency between Fig 5 and Table 2, 3. The situation is, for “shell,” “anime face,” and “pokemon,” the images shown in Fig 5 are already the best epoch from StyleGAN2, and they match the score in table 2 and 3. After that, more training leads to divergence. For the rest of the datasets, the quality increase from StyleGAN2 is also limited for longer training time. We have added a more detailed introduction to better relate the table 2,3 and figure 5 for the readers.
> >
> > 2.5: Our model is more robust from two aspects: 1. Consistency, and 2. Less mode collapse. It is an observation of all the tested datasets with different visual domains. For stylegan2 and baseline, they cannot converge or quickly to mode collapse on some tested datasets (very high FID, the generated images have clear visual defects). At the same time, our model performs consistently well on all datasets, and no mode-collapse is observed for even 20 hours of training.

---

> > > ### Comment · AnonReviewer2 · 2020-11-22
> > > **Response to point 2.1 and  2.2**
> > >
> > > I believe if the generator can only generate 100 training images this is a major concern, this means that the networks suffers from the memorization effect.  The network's only ability of reproducing the training set limits significantly its usability in practical applications.
> > >
> > > Thus I think there is a necessity of evaluating the proposed method with other evaluation metrics besides the FID score. For example, as proposed in my origina review to check the memorization effect one can measure LPIPS between the generated sample and its nearest training sample. Here the baseline for comparison would be to use the same metric but between the augmented training set (using the standard augmentations used for training FastGAN) and the original training set. If FastGAN produces the images which are more similar to the training set then the standards augmentations, then this is sign of the memorization effect and lack of diversity of the generated samples. In this case, I see very little use of FastGAN in practical applications.
> > >
> > > In our lab, we tried the code provided in the submission (using the default parameters). We trained FastGAN on the dataset of 100 images, and measured LPIPS (as well as MS-SSIM) as described above. In our experiment, FastGAN achieved lower LPIPS (LPIPS of 0.13) to the training set than the augmented version of the original dataset (LPIPS of 0.19). The same corresponds to the MS-SSIM metric (only higher here). By visual expection, it was also observed that FastGAN mostly reproduced the original training set. The quality of the generated images was good, which corresponded to a reasonable FID score.
> > >
> > > Taking into accounts our finding above, I think it's very important to compare the method using other metrics, besides FID. As in our experiments we didn't tune FastGAN to the used dataset (thus the results reported above might be unfair to the method), I would be very interested to see the above metrics on the reported datasets, e.g. such as Skull or Shell, using the exact model that the authors used for reporting the results in the paper.

---

> > > > ### Comment · AnonReviewer2 · 2020-11-22
> > > > **Thanks for your response**
> > > >
> > > > Overall, I would like to thank the authors for their detailed response and provided clarifications. Unfortunately, some of my major concerns have not been fully addressed. I think some important comparisons to the related work are missing in the paper, which would help to assess the effectiveness of the proposed technical contributions (SLE module and SS discriminator).
> > > >
> > > > Moreover, I'm also concerned with the potential memorization effect of FastGAN, thus I believe that additional metrics are needed (besides FID) to assess the diversity of the generated images and their difference from the training set. I would be very happy to upgrade my original score if the authors provide additional evaluations for the above mentioned point, showing that FastGAN produces images substantially different from the training set (running evaluations using LPIPS or MS-SSIM should take very little time and effort).

---

> > > > ### Author Response · Authors · 2020-11-22
> > > > **Experiments on LPIPS**
> > > >
> > > >
> > > > Regarding the LPIPS score. We will calculate the LPIPS on the nearest neighbor, and also show qualitative results on the generated images and their nearest neighbors.
> > > >
> > > > For the baseline that reviewer proposed, please let us know what exactly are the augmentations? Are they normal rescaling-and-cropping, or are they padding-and-cropping and also color-shifting (whitened, darken, contrast shifting) like what proposed in Zhao et al?

---

> > > > > ### Comment · AnonReviewer2 · 2020-11-22
> > > > > **Augmentations**
> > > > >
> > > > > Standard augmentations, e.g. cropping, flipping, etc. (no color-shifting).

---

> > > > > > ### Author Response · Authors · 2020-11-22
> > > > > > **Thanks**
> > > > > >
> > > > > > Thanks. As we are doing experiments, we are not sure if we can catch up with the close date of the revision submission, with only 2 days left.  I promise to include a more comprehensive comparison of the reviewer raised points of more datasets given more time. However, before the revision deadline, we may only able to cover a few datasets. We hope the reviewer can understand.

---

> > ### Comment · AnonReviewer2 · 2020-11-22
> > **Response**
> >
> > I have to disagree with the above comments. The paper claims two technical contributions:  SLE module and SS discriminator. However, for both proposed techniques the comparisons with the closest related works are missing. Thus it is very hard to assess the contributions.
> >
> > For SS-discriminator Chen et al. CVPR 2019 "Self-Supervised GANs via Auxiliary Rotation Loss" proposed to use rotation loss for self-supervision of the discriminator, so in contrast to the author's claim ("The idea is about integrating “self-supervision” to GAN, which no prior work has dedicated this idea to our knowledge") there is a prior work.  Moreover, Chen et al. showed that the proposed method works quite well for natural images (such as ImageNet), and I don't see a big problem in comparing to this method on rotation-sensitive datasets used in this submission, such as Anime Face or Art Paintings (which is used in Table 6 for comparison).
> >
> > Zhao et al. 2020 is the reference in this submission, i.e. "Differentiable augmentation for data-efficient gan training" arXiv'20, which also targets the problem of the overfitting of the discriminator when trained in a low-data regime.

---

> > > ### Author Response · Authors · 2020-11-22
> > > **Comparison to related work**
> > >
> > > Thanks for the reviewer's response.
> > > We will add a comparison with Chen et al and update the reviewer.
> > >
> > > TL;DR
> > > While running the new experiments, we would like to say:
> > > 1. Chen's method is about rotating the images, it is another self-supervised (ss) method that can cooperate together with our proposed auto-encoding. As we showed in the paper, these ss methods (contrastive, aspect-ratio prediction, auto-encoding) are not necessarily competitors, in some cases, they can work together and boost the performance (such as contrastive+aspect-ratio prediction ). Using these ss methods are more of a design choice, not counterparts. We should cite Chen et al and rephrase our contribution as proposing auto-encoding as an ss D, and we showed that it works better than without it.
> > >
> > > 2.  Rotating has limited using scenario compared to auto-encoding. In cases when rotating is not applicable, auto-encoding still can be used. This factor also strengthens our contribution.
> > >
> > > 3. For Zhao et al's method, we already used the method on all our experiments, including StyleGAN2, baseline model, and our model. Again, we do not think it is a counterpart. Zhao's differentiable data augmentation and our proposed ss discriminator work together and can achieve better performance than only using data augmentation alone (which is the baseline model).

---

> ### Author Response · Authors · 2020-11-22
> **Updated experiment results**
>
> Dear reviewer #2
>
> #
> update: we also upload the code and the trained checkpoints on many datasets, in the same link with our uploaded datasets. We are trying our best to increase the transparency of the methods in the paper.
> #
>
> Regarding the memorizing concern and the LPIPS score, we have quickly tested on various datasets and post the results here.
> Please check Appendix C, D, E, especially the new Figures 5 and 6, and Table 2. We show the randomly generated images from our model, and their closest real images according to the LPIPS score.
>
> We also compute the LPIPS score and show them below. We follow the detail provided by the reviewer: for the baseline, as the reviewer suggested, we compute LPIPS between each real image and a randomly augmented image of itself (random cropping and horizontal flipping); for our G, we random generate 100 images, and find the closest real image ranked by LPIPS, then compute the mean of all paired LPIPS values.  we run each computing 3 times and list all of them:
>
> LPIPS on nearest neighbor
>
> |                          | Art     |        |        |
>
> | training-data |  0.5529 | 0.5499 | 0.5518 |
>
> | Our-G         |   0.637 | 0.6572 | 0.6468 |
>
> |               | FFHQ-1k |        |        |
>
> | training-data |  0.5281 | 0.5283 | 0.5279 |
>
> | Our G         |  0.5859 | 0.5668 | 0.5982 |
>
> |               | Skull   |        |        |
>
> | training-data |   0.389 | 0.3957 | 0.3947 |
>
> | Our G         |  0.3168 | 0.3257 | 0.3228 |
>
> |               | Cat     |        |        |
>
> | training-data |  0.3903 | 0.3909 | 0.3898 |
>
> | Our G         |  0.5515 | 0.5622 | 0.5486 |
>
> |               | Dog     |        |        |
>
> | training-data |  0.3847 | 0.3896 |  0.396 |
>
> | Our G         |  0.5822 | 0.5647 | 0.5796 |
>
> |               | Shell   |        |        |
>
> | training-data |  0.3925 | 0.3847 | 0.3853 |
>
> | Our G         |  0.4275 | 0.4452 | 0.4296 |
>
>
> For the datasets, art and FFHQ are using exactly 1000 real images, and the cat, dog, skull, and shell all have sub-one hundred images. Our model has a higher LPIPS on all datasets except the skull. But for the skull, the synthesized images are actually not strongly imitating the real images, there are differences between them, so we show the images in Fig. 6 in the appendix for the readers to better perceive the results qualitatively. We will find a way to add this new table in the paper (currently there is no space left, we may re-arrange the paper to better sort the contents).
>
> We would like to thank you again for the reviewer's help and time. Please let us know if you are satisfied with the new results and if we are on the right track, as the discussion period is almost end.

---

> > ### Author Response · Authors · 2020-11-22
> > **Comparing to rotating self-supervision**
> >
> > Dear reviewer #2
> >
> > We have updated our related-work section to introduce the self-supervised D by citing Chen et al. We also added their "rotating" method in the experiment section.
> >
> > We have tried the rotation self-supervision on Discriminator. However, the model is unable to converge on both FFHQ-1k and Art-paintings-1k. It seems that this rotating signal for D severely hurt the learning of the whole GAN model. We tried to lower the weight for the rotating loss (from 1 to 0.5, 0.2), and the model performs no better than the baseline, with a clear visual quality difference to our model (the generated images from the model with rotating ss are defects and noisy color patches). The FID of rotating with weight 1 is 83 on art and 57 on ffhq-1k, both severally lower than baseline.
> >
> > This observation makes us doubt the rotating's performance on few-sample datasets. just as we showed in our paper, contrastive and aspect-ratio also do not perform as well in our low computing budget settings. In Chen's paper, see Table.1, the rotating performance on CelebA is even worse than unconditioned GAN. As we hypothesized in our paper, except for auto-encoding, other ss methods could lead the model to focus on local regions to find cues for making predictions (class label, asp-ratio, rotate degree) and lead to overfitting, and such behavior is only more sever when training data is limited. In contrast, auto-encoding always forces the model to pay attention to all spatial regions of an image.
> >
> > According to Chen's provided code, they train the model only on 128 resolution, and on full datasets with more than 10k images, and with a big batch size 64. It means their study focus is different from ours, and their observations can not transfer to our study settings. Our paper is focused on a limited computing budget and training images, but higher resolutions (less than 1k images, batch size of 8, 1024 resolution images). This factor further makes it hard for us to compare rotating and auto-encoding.
> >
> > Importantly, we want to emphasize that: we provide 2 methods that actually improve the performance than without them, we support the claim with experiments comparing to the best prior works to our knowledge and provided code and datasets. Both "SLE" and "auto-encoding as self-supervision" are not been proposed before, and even the study topic itself (faster GAN on smaller datasets with limited GPU power) is new and rarely studied.
> >
> > Hope our provided new results could dispel some worries from the reviewer. We appreciate the detailed and patient discussions from the reviewer, and thank you again for your help!

---

> > > ### Comment · AnonReviewer2 · 2020-11-24
> > > **Thanks for the prompt response**
> > >
> > > Dear authors,
> > >
> > > Thanks for the provided extra evaluations and clarifications. I feel that my concerns have been adequately addressed and I am revising my score to a 7.

---

### Official Review · AnonReviewer1 · 2020-10-28

**Rating:** 7
**Confidence:** 3

**Review:**

Summary:
This paper introduces a new GAN architecture that targets high resolution generation for small datasets. Two techniques are introduced for this purpose: skip-layer channel-wise excitation (SLE) modules, and regularization of the discriminator via a self-supervised auxiliary task. The proposed architecture is shown to outperform current SOTA models on a variety of small datasets, while training in less time.

Strengths:
-Paper is well written and easy to understand.
-SLE combines benefits of skip-connections, channel-attention, and style-modulation in a single operation.
-Self-supervised discriminator appears to be very effective at preventing overfitting in low data regimes.
-Strong generation results on a variety of datasets, including better image quality, and faster training time than the baseline models with similar numbers of parameters.
-Ablation study demonstrates the usefulness of each of the proposed components.

Weaknesses:
-No significant weaknesses that I can think of.

Recommendation and Justification:
I quite like this paper and tend to vote for acceptance. It is refreshing to see a new architecture designed specifically for the low data, low compute regime, rather than simply reducing the capacity of existing architectures. I particularly like the idea of regularizing the discriminator with an auto-encoding task. Many other methods that attempt to combine auto-encoding and GANs seem to constrain the model too much due to the requirement of mapping all examples in the dataset into the latent space, but this method does not appear to share this constraint. It also has the added benefit of sharing discriminator capacity, rather than introducing an additional encoder which further increases computational cost.

Clarifying Questions:
-Why perform random cropping in the discriminator at 16x16 resolution? Why not perform reconstruction on the full image? Is this mainly for computational savings?
-Is there any weighting on the reconstruction loss in Equation 3 or is the weighting effectively equal to 1 here?
-In Table 4, the full StyleGAN2 model outperforms the proposed model when more images are available. However, as is stated in the paper, the StyleGAN2 model has twice as many parameters. If the number of parameters in both models were equal (either by doubling the amount in the proposed model of halving the amount in the StyleGAN2), which would be expected to achieve better performance?

---

### Official Review · AnonReviewer4 · 2020-10-28
**Tend to Accept**

**Rating:** 7
**Confidence:** 4

**Review:**

Summary:
This paper proposes a lightweight GAN architecture which is tuned for learning generative models in the case where one has access to only a relatively small datasets, as well as a simple autoencoding modification for GAN discriminators to help prevent overfitting and mode collapse. Results are presented on a range of benchmark and new datasets, using standard metrics to compare performance against existing models. The new models compare favorably in the target regime, while unsurprisingly not being as strong as the baseline models in the large-data regime.

My take:

This is a decent paper, with reasonable (albeit not perfect) presentation clarity and passably significant results. The architectural modifications are not trivial (i.e. one could just try and make a StyleGAN less wide, but the authors give specific attention to the design of the higher resolution layers in G), and the changes to the training procedure are simple and effective, while being sufficiently different from previous autoencoder-based approaches to merit being called novel. The results show that the proposed models perform well (qualitatively and quantitatively) in the low-data regime against a full-size StyleGAN baseline, particularly when controlling for compute budget. I would rate this paper about a 6.5: I do not have any major concerns (I would evaluate the technical and methodological soundness of this paper as high) but I also would not expect this paper to have an especially high impact. As I tend to accept, I expect to reconsider my rating to a 7 after the discussion period unless major concerns are raised.

My main suggestion to the authors which I think could strengthen this paper would be to compare against a baseline StyleGAN with the width multipliers decreased. If I were a practitioner seeking to improve model performance in the low-data regime, this would be my first approach: to take an existing, working model, and make it smaller. As the authors’ architectural changes are not this simple, one would expect that, for an equivalent FLOP budget and training budget, the new architecture would outperform a “StyleGAN-slim,” but it would be good to have quantitative evidence on this front.

If the authors have the compute budget, it would also be good to see how this model “scales up,” in the case where it is made deeper or wider (simply changing the width multipliers) and tested against StyleGAN on the full FFHQ; a plot comparing FID over time for the two models (so that a practitioner could see how long one would have to train an equivalent StyleGAN on the full dataset to outperform this model, or vice versa) would be useful. Since the model has a different architecture I would expect it to have different scaling properties. However, this reviewer appreciates that this would be a compute intensive experiment that is likely not possible to run in the revision period, and does not wish to push the authors in this direction given that it is outside the target scope of low-data modeling.


“Note that we collect the last six datasets in the wild, which we do not have a license to re-distribute, yet one can freely collect them just like us.”

I appreciate that the authors have given some considered the legal implications of distributing potentially copyrighted images (and given that there’s not much established legal precedent that I’m aware of on whether doing research using copyrighted images of Pokemon constitutes fair us, this reviewer does not consider this cause for concern). For open-sourcing, the authors might want to consider releasing the URLs of the datasets to enable reproducibility (at least for as long as the images are up), which is an approach that has been used for other datasets. This would also allow the authors revise this sentence to be a bit more “academic,” e.g. “Note that we do not have a license to re-distribute the last six datasets, which we collect from the wild, but we provide the URLs in order to enable reproducibility.”

Edit: As mentioned in my comment below, I believe the authors have done sufficient work in their revision to address my concerns and am revising my score to an acceptance.

---

> ### Public Comment · ~Gwern_Branwen1 · 2020-11-16
> **Anime faces are PD**
>
> re dataset copyrights: if it helps, judging from the StyleGAN artifacts, I suspect the anime faces in this paper are sourced from my ThisWaifuDoesNotExist, in which case they are public domain because they were randomly generated, and there are no concerns about redistributing in any form. Incidentally,  the original dataset of 300k real anime faces used to train the TWDNE models is already available for download (https://www.gwern.net/Crops#danbooru2019-portraits) if they want to investigate sample size scaling.

---

> > ### Author Response · Authors · 2020-11-16
> > **thanks**
> >
> > Thanks for your comment. And yes, we believe all images we used can easily be found online. We will pack the exact portion of images we used and upload them to google drive for easier access. Currently, we are trying to include more datasets that cover larger varied domains.

---

### Author Response · Authors · 2020-11-16
**training data and code now available**

We have made the data, the code, and the trained checkpoints available from this link: https://drive.google.com/drive/folders/1nCpr84nKkrs9-aVMET5h8gqFbUYJRPLR?usp=sharing
It should be an anonymous link, and our identity should not be revealed.

Our code is already submitted in the supplementary material, including both baseline config and the proposed two techniques. One can easily run our code on the provided image sets and customized images.

---

### Decision · Program_Chairs · 2021-01-07
**Final Decision**

**Decision:**

Accept (Poster)

**Comment:**

The paper proposes a method for training GANs in few-shot setting. Two key components of the method are: a skip-layer channel-wise excitation (SLE) module that encourages gradient flow across resolutions, and a self-supervised loss of autoencoding to regularize the discriminator. The results presented in the paper are indeed impressive in the few-shot setting. Reviewers had some concerns about training set memorization which have been addressed by the authors with additional evaluations using LPIPS metric. Overall, the paper tackles an important problem of few-shot learning of GANs and will be a good addition to the ICLR program.